# Pisa Syndrome Secondary to Drugs: A Scope Review

**DOI:** 10.3390/geriatrics9040100

**Published:** 2024-07-30

**Authors:** Jamir Pitton Rissardo, Nilofar Murtaza Vora, Naseeb Danaf, Saivignesh Ramesh, Sanobar Shariff, Ana Letícia Fornari Caprara

**Affiliations:** 1Neurology Department, Cooper University Hospital, Camden, NJ 08103, USA; ana.leticia.fornari@gmail.com; 2Medicine Department, Terna Speciality Hospital and Research Centre, Navi Mumbai 400706, India; nilofar031202@gmail.com (N.M.V.); ramyaramesh2706@gmail.com (S.R.); 3Medicine Department, Lebanese University, Hadath RGHC+4PR, Lebanon; naseeb.danaf@st.ul.edu.lb; 4Faculty of General Medicine, Yerevan State Medical University, Yerevan 0025, Armenia; sanobarshariff@gmail.com

**Keywords:** Pisa syndrome, pleurothotonus, dystonia, tardive dyskinesia, extrapyramidal symptom, drug induced, movement disorder

## Abstract

Background: Pisa syndrome, also known as pleurothotonus, is a neurological condition characterized by more than ten degrees of constant lateral curvature of the spine when upright. In this way, the present manuscript aims to systematically review Pisa syndrome secondary to drugs. Methods: Two reviewers identified and assessed relevant reports in six databases without language restriction between January 1990 and June 2024. Results: The prevalence of Pisa syndrome varied from 0.037 to 9.3%. We found 109 articles containing 191 cases of drug-induced Pisa syndrome reported in the literature. The mean and median ages were 59.70 (SD = 19.02) and 67 (range = 12–98 years). The most prevalent sex was female, 56.91% (107/188). The most frequent medications associated with Pisa syndrome were acetylcholinesterase inhibitors in 87 individuals. Of 112 individuals in which the onset time from the medication to the movement disorder occurrence was reported, 59 took place within a month. In this way, a return to baseline was observed in 45.50% of the cases, and partial recovery was observed in 14.28%. Conclusion: We proposed new diagnostic criteria for Pisa syndrome based on previous findings in the literature. Moreover, multiple mechanisms are probably involved in balance control and the development of lateral trunk flexions.

## 1. Introduction

Pisa syndrome in humans, also known as pleurothotonus, is a rare condition characterized by more than ten degrees of constant lateral curvature of the spine when upright, without any evident rotation of the spinal bones, resembling the posture of the Leaning Tower of Pisa, after which it was named (Figure 1). This disorder is rare and can arise from the extended use of medications such as antipsychotic drugs or from other neurological conditions [1]. Many clinicians lack familiarity with this syndrome, which often contributes to delays in diagnosis, a decreased quality of life, and unnecessary diagnostic tests and treatments [2]. Pisa syndrome was first illustrated by Ekbom et al. in 1972, yet it was noticed more frequently in females with neurological conditions who were medicated with conventional antipsychotics [3].

The pathophysiological mechanisms underlying Pisa syndrome are still unclear. Some researchers suggest that it is related to changes in the neural circuits that control body posture and movement. Others propose that this syndrome occurs due to anatomical deformities in the muscles and bones. Moreover, some medications, especially those used for the management of Parkinson’s disease, can trigger Pisa syndrome [4]. In this context, Pisa syndrome could present as a rare side effect of various drugs from many different classes, mainly antipsychotics, dopaminergic agents, and cholinesterase inhibitors [5]. The exact mechanisms are still unknown. However, one of the most widely accepted hypotheses is that Pisa syndrome results from a cholinergic–dopaminergic disbalance induced by antipsychotic therapy. The aging effect, female sex, central nervous system diseases, and polypharmacy are strongly associated with the development of Pisa syndrome [6].

Pisa syndrome is a condition that affects the posture and balance of the body, making it lean to one side. It can cause imbalance, discomfort, and impairment in the ability to walk. It can also affect the quality of life and self-esteem of patients. Therefore, promptly diagnosing this condition can lead to adequate management and preventing unnecessary therapies [6]. Considering that individuals with Pisa syndrome can sometimes achieve significant improvement or even resolution of their signs and symptoms, clinicians must learn how to diagnose and manage this syndrome. However, more studies are needed to find the best ways to manage this condition [5,6].

The present study aims to systematically review Pisa syndrome secondary to drugs, its clinical presentation, duration, diagnostic assessments, and treatments.

## 2. Methodology

### 2.1. Search Strategy

For this systematic review, we searched six databases to locate existing reports of Pisa syndrome secondary to drugs published from January 1990 until June 2024 in electronic form. Excerpta Medica (Embase), Google Scholar, Latin American and Caribbean Health Sciences Literature (Lilacs), Medline, Scientific Electronic Library Online (Scielo), and ScienceDirect were searched. Search terms were “pisa syndrome” and “pleurothotonus” (Table 1). Preferred Reporting Items for Systematic Reviews and Meta-Analyses (PRISMA) 2020 checklist was followed [7].

### 2.2. Inclusion and Exclusion Criteria

Case reports, case series, original articles, letters to the editor, bulletins, and poster presentations published from January 1990 to June 2024, without language restriction to ensure a thorough review. In the cases where the non-English literature was beyond the authors’ proficiency (English, French, and Spanish) or when the English abstract did not provide enough data, such as articles in Chinese, Japanese, and Korean, Google Translate service was used [8].

The authors independently screened the titles and abstracts of all articles from the initial search. Disagreements between authors were solved through discussion. Cases where the cause of MD was already known and the motor symptoms either did not worsen or were unrelated to medication were excluded. Also, cases not accessible by electronic methods, including after a formal request to the authors, were excluded. Cases with more than one factor contributing to the MD were evaluated based on the probability of the event occurrence based on the Naranjo algorithm.

### 2.3. Data Extraction

A total of 5527 articles were found; 2207 were inappropriate, and 3211 were unrelated to the subject, duplicates, inaccessible electronically, or provided insufficient data (Figure 2). Data abstraction was performed. When provided, we extracted author, department, year of publication, country of occurrence, number of patients affected, cause of Pisa syndrome, time from first use of the drug until the Pisa syndrome onset, management of the Pisa syndrome, and patient’s status at follow-up, and significant findings of clinical history and management. The data were extracted by two independent authors and double-checked to ensure matching.

### 2.4. Statistical Analysis

Categorical variables were represented as proportions. Continuous variables were represented as means, standard deviation (SD), median, and range. Statistical analysis was performed using Microsoft Excel Spreadsheet Software version 16.0 (Microsoft Corp, Redmond, WA, USA).

### 2.5. Definitions

The clinical characteristics and definitions of Pisa syndrome were obtained from Ekbom et al. [3]. The Naranjo algorithm was used to determine the likelihood of whether an adverse drug reaction was actually due to the drug rather than the result of other factors [9]. Movement disorder onset was defined as the time from the medication start until the development of the movement disorder. Movement disorder recovery was defined as the time from the first management, which could be the medication discontinuation, until the full recovery of the abnormal movement [10].

## 3. Prevalence and Risk Factors Associated with Pisa Syndrome

Pisa syndrome is an uncommon adverse reaction to drugs (Table 2). The prevalence of Pisa syndrome is not well known because of the lack of clear diagnostic criteria and the paucity of epidemiological large-scale studies [10]. This condition is believed to affect predominantly the female sex. Also, it is frequently overlooked and mistaken for other syndromes and conditions. Moreover, it is more prevalent in young and elderly females with an organic central nervous system pathology. Interestingly, this feature is different from other dystonic reactions secondary to drugs, in which the most common age category affected is middle-aged individuals [11].

Pisa syndrome often improves upon the discontinuation of offending drugs or when its daily dose is reduced, but some patients may experience worsening symptoms over time. One of the available treatments for Pisa syndrome involves the prescription of anticholinergics, which are effective in about forty percent of the patients [18]. Interestingly, the response to treatment in Pisa syndrome is better when compared to tardive dystonia but inferior to that of acute dystonia. In this context, the underlying mechanisms of Pisa syndrome secondary to drugs are not fully understood. Still, they may generally involve a disbalance between dopamine and acetylcholine levels or a dysfunction of the serotonin and norepinephrine systems.

In patients with Parkinson’s disease and atypical forms of parkinsonism, an array of postural deformities may manifest, notably including camptocormia, anterocollis, retrocollis, and scoliosis. The lateral flexion of the trunk in patients with Parkinson’s disease was historically described as “the scoliosis of parkinsonism [19]”. These abnormal postures can reduce the quality of life and cause significant disabilities in people with Parkinson’s disease. Although postural abnormalities are more frequent in Parkinson’s disease, a stooped posture with flexion of the knees and trunk, bent elbows, and arms, which are adducted and flexed, are the most common manifestations [20].

Camptocormia is distinguished by a pronounced forward flexion of the thoracolumbar spine exceeding 45 degrees in the sagittal plane during ambulation, a phenomenon markedly reversible in the recumbent position [21]. In this context, anterocollis involves a severe anterior inclination of the head surpassing 45 degrees in the sagittal plane, partially reversible through voluntary movement but not entirely counteracted by gravitational forces. While infrequent in Parkinson’s disease, anterocollis is prevalent in other forms of parkinsonism, such as multiple system atrophy [22]. On the other hand, retrocollis manifests as a posterior positioning of the neck in the sagittal plane, an uncommon occurrence in Parkinson’s disease but characteristic of progressive supranuclear palsy [23]. Also, patients with Parkinson’s disease can develop scoliosis, characterized by lateral curvature exceeding ten degrees in the coronal plane, as assessed by the Cobb method. Additionally, patients with scoliosis often resist both voluntary and passive rectification, concurrently exhibiting axial torsion, which can be evaluated in a panoramic radiograph image of the spine [24].

Notably, the flexible deformity of Pisa syndrome differs from the inflexible deformity associated with scoliosis. The latter is diagnosed by lateral bending with a rotation of the vertebral bones [23]. Pisa syndrome in Parkinson’s disease may have multiple causes, such as asymmetric basal ganglia dysfunction, impaired sensory processing, cognitive impairment, and altered body alignment perception [11]. The presence of comorbidities such as osteoporosis and osteoarthritis may also increase the likelihood of developing postural abnormalities with low treatment response over time (structured Pisa syndrome) [5]. The interaction of complex central (dopaminergic and non-dopaminergic) and peripheral mechanisms may account for the diverse clinical presentation and prognosis of Pisa syndrome in Parkinson’s disease and other categories of parkinsonism [5]. The patient’s posture should not be confused with a functional neurological disorder or catatonic state. The cause of Pisa syndrome induced by drugs is not clearly understood. Interestingly, Pisa syndrome that does not respond to anticholinergic medications may be due to permanent brain damage or peripheral causes [25]. Idiopathic Pisa syndrome, which is not caused by drugs, is a rare condition that affects the trunk muscles in adults and is non-related to antipsychotics. The most frequently reported treatment for idiopathic Pisa syndrome is high doses of anticholinergic medications [18].

Pisa syndrome is considered a form of dystonia by some movement disorders specialists. It can be either acute or tardive, depending on the onset and duration of the symptoms. Also, some authors characterize it as tardive dyskinesia; the term “tardive” means that the symptoms appear late or after a long period of exposure to the medication. The term “dyskinesia” means abnormal or impaired movement. Pisa syndrome usually responds well to anticholinergic treatment [26]. However, some patients may have a refractory form of Pisa syndrome that may share pathophysiological features with tardive dystonia. A critical difference between the two conditions is the outcome of decreasing the dosage or discontinuing the antipsychotic medication. Pisa syndrome tends to improve markedly, while tardive dystonia remains persistent. In this way, Pisa syndrome is reversible, while tardive dystonia is irreversible [18].

Furthermore, clinical management of Pisa syndrome is usually based on empirical evidence because no specific drugs have been developed. In this context, the first step in the management of this syndrome is to withdraw the offending medication or, for the mild cases, to decrease dosages [27]. The second step is to consider initiating anticholinergic therapy. Güneş et al. reported a case of risperidone dose tapering after the development of Pisa syndrome, with the complete recovery of motor symptoms. Also, the authors discussed a possible relationship between extrapyramidal symptoms and the development of Pisa syndrome [28].

### Risk Factors

Exposure to antipsychotic medications is a significant risk factor for the development of Pisa syndrome. It is believed that the imbalance between dopamine and acetylcholine in the basal ganglia may be the cause of the variable rates of dystonia associated with various antipsychotic medications [10]. Acute dystonia is more likely to occur with a larger ratio of dopamine–acetylcholine antagonists [29]. One of the hypotheses for the effectiveness of low-potency antipsychotic medications in reducing the incidence of acute dystonia is their anticholinergic effects [10]. Because an initially small dose of antipsychotic is capable of decreasing the risk of acute dystonia compared to routinely prescribed doses, it is recommended that the lowest effective dose of high-potency antipsychotic medications is started [30]. In this context, it is hypothesized that low-potency antipsychotics are less frequently associated with Pisa syndrome.

A prospective study showed that cocaine use was associated with an increased incidence of Pisa syndrome; moreover, patients with parkinsonism and dementia appear to be at higher risk of developing Pisa syndrome secondary to drugs [5]. Cocaine can temporarily increase the levels of serotonin and norepinephrine by inhibiting their reuptake. In this way, mild parkinsonian signs could be explained by the long-term use of neuroleptics in patients without a history of previous neurological conditions or non-motor symptoms [30]. Interestingly, cocaine users have an increased risk of developing dystonic reactions when they use antipsychotic medications [31]. Mascia et al. reported a case showing that long-term neuropsychiatric drugs can increase the risk of acute Pisa syndrome caused by cocaine. The combination of acute cocaine use and chronic psychiatric treatments may predispose to a complex imbalance of neurotransmitters, leading to acute muscle spasms affecting the trunk [31].

## 4. Drugs Associated with Pisa Syndrome

Pisa syndrome is predominantly caused by the long-term use or high dosage of antipsychotic drugs. Although antipsychotic drugs are known to be the main culprits in this syndrome, several other drugs are reported to be associated with this syndrome as well [32]. Additionally, antidepressants, psychoactive drugs, and antiemetics have also been found to be associated with Pisa syndrome [33,34]. In this review, we found 109 articles in the literature reporting 191 cases of drug-induced Pisa syndrome (Table 3). The mean and median ages were 59.70 (SD = 19.02) and 67 (range = 12–98 years). The most prevalent sex was female, 56.91% (107/188).

The most frequent medications associated with Pisa syndrome were acetylcholinesterase inhibitors in 87 individuals. The other medications were 50 atypical antipsychotics, 30 antipsychotics, 6 antiparkinsonian medications, 4 anti-seizure medications, 3 antidepressants, 3 mood stabilizers, 3 tricyclic-antidepressants, 1 opioid, 1 antiemetic, 1 addictive stimulant drug, 1 intoxication, and 1 anti-vertigo medicine (Table 4). Of 112 individuals in which the onset time from the medication to the movement disorder occurrence was reported, 59 occurred within a month. Complete recovery was reported in 86 individuals, and clinical improvement occurred within 3 months in 82 individuals. In this way, a return to baseline was observed in 45.50% of the cases, and a partial recovery was observed in 14.28%. However, 40.22% did not report clinical outcomes regarding improving the movement disorder.

## 5. Physical Examination and Diagnostic Criteria

The lateral displacement of the trunk and initial presentation of postural misalignment reported by patients and their caregivers are the basis for the clinical diagnosis of Pisa syndrome. Assessing the trunk’s misalignment can be easily achieved by using a wall goniometer, an inclinometer, or even smartphone apps [132]. The best way to determine the angle of curvature in the coronal and sagittal planes according to the Cobb angle is to take a radiograph of the patient’s spine while standing [133].

Tilting symptoms occurring bilaterally may be labeled as “metronome or alternating Pisa syndrome [44]”. All body parts should be carefully inspected when the patient has been exposed properly. A radiograph should be taken in both the standing and supine positions to assess vertebral rotation and rule out structural bone abnormalities and scoliosis [134].

Although the diagnostic standards for Pisa syndrome are not agreed upon, at least ten degrees of lateral flexion should be considered (Table 5). Bonanni et al. defined Pisa syndrome as a lateral flexion of the trunk greater than fifteen degrees, which should not be observed in the supine position and which increases during walking and in the absence of any mechanical restriction movement of the trunk with continuous electromyographic activity in the lumbar paraspinal muscles ipsilateral to the side of the bend [135]. On the other hand, Doherty et al. recommend that a diagnosis of Pisa syndrome requires severe lateral flexion, which they described as at least 10º of lateral flexion that can be completely alleviated by passive mobilization or lying in a supine position. Also, the authors included that Pisa syndrome can be defined without electrophysiological studies [24].

It is worth mentioning that scoliosis could also occur in the presence of Pisa syndrome. One distinguishing clinical hint is that scoliosis is generally not reversible by the change in the spinal curvature when the patient is lying down [137]. Even though there are several documented causes of spinal deformity in adults, Pisa syndrome in Parkinson’s disease may occasionally coexist with scoliosis. Doherty et al. proposed a lateral flexion of at least 10 degrees as a diagnostic criterion, which can be entirely reversed by passive mobilization or supine posture [136]. 

The angle of the lateral trunk flexion can be used to classify Pisa syndrome as mild (<20°) or severe (>20°) [14]. Although these definitions are generally used in clinical practice, no consensus on the clinical definition or diagnostic criteria of Pisa syndrome has been established.

An electromyographic study of the activation pattern of paraspinal muscles like the longissimus muscle and non-paraspinal muscles like the external oblique muscle during different postures might provide an additional diagnostic criterion. Continuous electromyographic activity of the lumbar paraspinal muscles ipsilateral to the bending side while standing or walking has been hypothesized to provide further diagnostic data [133,135,138].

## 6. Pathophysiology

The dopaminergic system is thought to be involved in developing Pisa syndrome. Animal studies have revealed a deviation in favor of the denervated side when performing nigrostriatal lesions [139,140,141,142]. Patients with unilateral stereotactic subthalamotomy that deviate towards the opposite side of the lesion have reported seeing similar observations [142]. In these studies, the lateral tilt was eliminated entirely after a contralateral subthalamotomy and rectified, at least in part, with levodopa. However, multiple mechanisms are probably involved in balance control and the development of lateral trunk flexions (Figure 3) [15,38,69,97,143,144,145,146,147,148].

Non-dopaminergic pathways may be involved in developing Pisa syndrome. Postural control is associated with the cholinergic system in the striatum and the pedunculopontine nucleus [149]. An asymmetric input to the pedunculopontine nucleus could be responsible for a trunk muscle tone imbalance. It could also explain posture normalization during the supine position because of the decreased antigravity extensor muscle tone. Supporting findings of this hypothesis could be the lack of improvement with dopaminergic medications [150]. Another fact is the strong relationship between Pisa syndrome and the improvement with cholinesterase inhibitors. Also, deep brain stimulation in the left pedunculopontine nucleus showed improvement in the motor manifestations of Pisa syndrome [151]. In this way, these findings may suggest the role of the cholinergic system in developing Pisa syndrome. 

**Figure 3 geriatrics-09-00100-f003:**
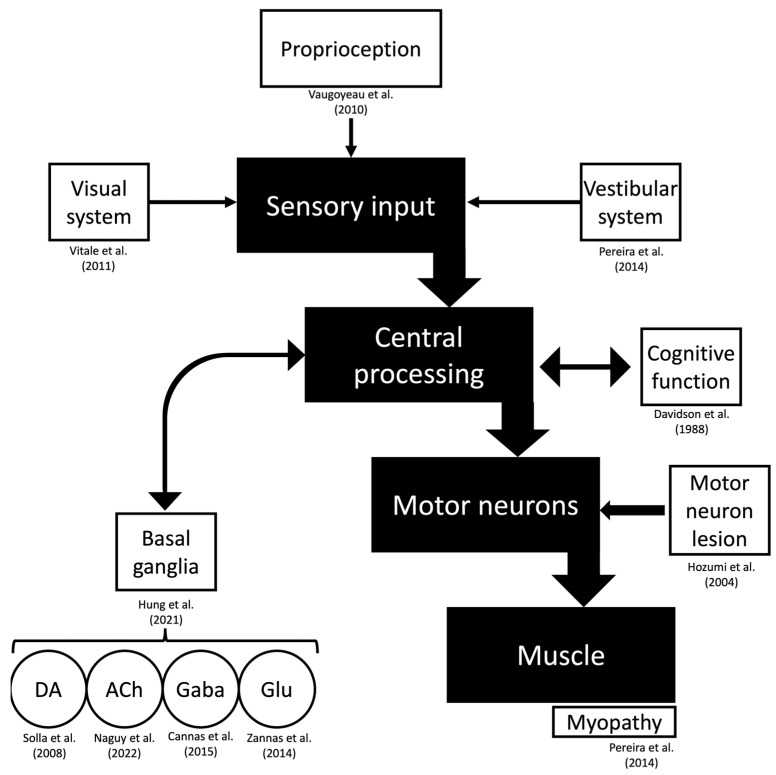
Pathophysiology of drug-induced Pisa syndrome. The motor function is represented, and possible hypotheses from the literature regarding specific structures of this pathway are referenced. Abbreviations: ACh, acetylcholine; DA, dopamine; Gaba, Gamma-aminobutyric acid; GLU, glutamate [16,39,63,70,98,144,145,146,147,150].

A study assessing the FDA Adverse Event Reporting System for reported cases of Pisa syndrome related to medications found that fifty-two reports of this syndrome were linked to cholinesterase inhibitors. Also, the study described the same adjusted reporting ratios of Pisa syndrome with donepezil, rivastigmine, and galantamine. Additionally, they noted a strong correlation between the Pisa syndrome and the use of memantine, a drug whose mechanism of action involves the blockage of N-methyl-D-aspartate receptors, indicating that the development of the syndrome may not be limited to the dopaminergic-cholinergic system and may instead be caused by an imbalance in different neurotransmitters. The indications for the prescription of the cholinesterase inhibitors were Alzheimer’s disease (26 cases), cognitive impairment (3 cases), dementia with Lewy bodies (2 cases), vascular dementia (1 case), Huntington’s disease (2 cases), and depression (2 cases), while in 15 cases, the diagnosis was unknown. Notably, 25 patients were also on antipsychotic medications [97].

In animal models, methamphetamine and a combination of a dopamine reuptake inhibitor and galantamine lead to rotatory movements resembling Pisa syndrome in humans. Interestingly, the dopamine release was enhanced by anticholinesterase inhibitors in the contralateral side of the striatum in the hemiparkinsonian model in rats [141]. Interestingly, although there was no difference in the number of reports of Pisa syndrome associated with anticholinesterase inhibitors, studies in mice revealed a stronger association with galantamine than with donepezil [152]. It is believed that a weaker antagonistic effect against the muscarinic receptors is found in galantamine than in donepezil [142].

In one investigation on the regulation of postural orientation in Parkinson’s disease, somesthetic signals originating from sensory systems such as muscles, skin, and joints were assessed. The study participants engaged in an activity with their eyes closed while exposed to slow oscillations on a motorized rotating platform below the semicircular canal sensory threshold. In contrast to control participants, those with Parkinson’s disease had difficulty controlling their postural orientation based on sensory input, which suggests that proprioceptive information loss may play a role in the development of lateral trunk flexion [153]. Vitale et al. reported that visuospatial dysfunction assessed by Benton’s judgment line orientation test was a strong and significant predictor for developing Pisa syndrome [154]. Another possible explanation is a disruption in “body schema,” which is defined as a self-three-dimensional internal representation of the body biomechanics, instead of the visuospatial system since the cortical area found to be abnormal was the bilateral parietal lobe, which has a significant role in the “body schema” system [155]. Also, Biassoni et al. found that compared to controls, patients with Pisa syndrome have significant hypometabolism in bilateral temporal–parietal regions, mainly in Brodmann area 39 and the bilateral posterior cingulate cortex [156]. It is noteworthy that Yoritaka et al. showed that in individuals with Parkinson’s disease before the onset of Pisa syndrome, hypoperfusion in the correlative visual cortex and the position discrimination test are observed [157]. 

Recent research in individuals with Parkinson’s disease and lateral trunk flexion showed an impairment of vestibular function may be partially responsible for postural orientation abnormalities [158]. A peripheral, unilateral vestibular hypofunction was observed in individuals with lateral trunk flexion. Vestibular hypofunction was found to be ipsilateral to the leaning side and contralateral to the most severely affected Parkinsonian side [145]. Patients with Parkinson’s disease have a compromised perception of verticality and a disturbed processing of graviceptive pathways. This has also been correlated with scoring in the Unified Parkinson’s Disease Rating Scale and the Hoehn and Yahr Scale [24].

Scocco et al. examined Parkinson’s disease patients with and without Pisa syndrome and healthy controls, basing their comparisons on the premise that changes in the subjective visual vertical function may cause the anomalous posture of patients with Pisa syndrome [159]. They observed that all the groups had the same subjective visual verticality. Thus, the changes seen in Pisa syndrome were probably not related to the intrinsic lateral deviation. It is crucial to understand whether this is due to changes in the secondary or peripheral integration of otolithic imbalance or a different verticality depiction. Additionally, it has been proposed that a musculoskeletal system abnormality may cause the incorrect posture observed in patients with Parkinson’s disease. However, the evidence for this concept mostly comes from research on patients with camptocormia [21]. In one study, the authors examined the paraspinal muscles of 14 patients. They discovered nonspecific myopathic alterations, including the hypertrophy of type 1 fibers, the loss of type 2 fibers, a decrease in the activity of oxidative enzymes, and myofibrillar disorganization [146].

In another study, individuals with Pisa syndrome demonstrated difficulties engaging paraspinal muscles during the clinical test when asked to assume an upright position. The study observed the patients for six years after the onset of Pisa syndrome. The individuals would hyperextend their necks to seem taller or press down on one knee [24]. However, electromyographic studies were not performed. There was muscular hyperactivity but no denervation or myopathic pattern in electromyography studies in Pisa syndrome patients [133,160]. According to an electrophysiological survey, Pisa syndrome is characterized by either hyperactivity of the ipsilateral paraspinal lumbar muscles (pattern 1) or hyperactivity of the contralateral paraspinal muscles combined with ipsilateral hyperactivity of the non-paraspinal lateral trunk muscles (pattern 2) [138].

One possible hypothesis to explain the susceptibility of individuals with neuropsychiatric diseases to develop Pisa syndrome is the role of oxidative stress mechanisms, which contributes to the degeneration of dopaminergic neurons and, consequently, motor dysfunction. Also, a similar pathological mechanism can be presumed in individuals with a history of traumatic brain injury [128].

## 7. Clinical Manifestations

Pisa syndrome can emerge subacutely and deteriorate rapidly within a few months, or it might develop persistently with a subclinical start and slow progression [160]. When compared to individuals with chronic exposure, whose onset of the syndrome is gradual and less sensitive to anticholinergic medication, the latter presentations have been associated with recent exposure to antipsychotics and have benefitted the most from anticholinergic treatment [12,24,25]. Also, it has been suggested that these deficits worsen with the progression of the disease. Moreover, special attention should be given to these acute and subacute cases. There are several reports in the literature of this subgroup of individuals being misdiagnosed with functional disorders [43].

Drug-induced Pisa syndrome with subacute onset, in contrast to chronic forms of Pisa syndrome, is probably less associated with local changes affecting muscle and bones [4]. Also, electromyographic features in subacute onset cases revealed a plastic striatal change secondary to neurotransmitter dysregulation [160]. Moreover, a possible explanation for the predisposition of some individuals to develop Pisa syndrome could be a difference between the concentration of neurotransmitters in both hemispheres [18]. Additionally, individuals with intellectual disability have shown an increased risk of Pisa syndrome [81].

In individuals with Parkinson’s disease, it was reported that patients could lean towards or lean away from their predominantly PD-affected side [5]. However, since individuals can lean towards or away from the side most affected by Parkinson’s disease, other mechanisms, rather than basal ganglia asymmetry, could potentially contribute to the development of Pisa syndrome [121]. Also, there are reports of worsening of the lateral truncal flexion during the on-and-off periods in patients with Parkinson’s disease [68].

According to some authors, Pisa syndrome should be categorized as acute or as tardive according to the exposure to antipsychotics [161]. The duration between the medication onset and the development of Pisa syndrome should be taken into consideration for prognostication since patients who develop this syndrome subacutely will have an increased chance of achieving full recovery [129]. The therapeutic choice of changing or discontinuing particular drugs can assist in distinguishing between chronic processes with less likelihood of improvement. In patients with neurodegenerative conditions with low baseline motor function and degree of mobility, Pisa syndrome may first be observed as a propensity to lean to one side while seated in a chair, followed by lateral flexion when they walk [162]. Dyspnea or instability that can cause falls might occur as the deformity progresses, in addition to discomfort [159,163]. During a physical examination, patients may unintentionally lean to one side while sitting, standing, or walking. It is interesting to note that patients frequently fail to notice the tilt, and it has been suggested that this is because their sense of vertical and horizontal locations has changed [163,164].

Some authors consider Pisa syndrome as a subtype of dystonia [25]. Due to the static, neither fixed nor movable posture, the lack of overflow, twisting, or twitching, and the lack of sensory ticks, it has been questioned whether Pisa syndrome correlates to dystonia [7]. However, even though there are no anatomical abnormalities, the quick onset of the disease in specific individuals and the absence of other symptoms point to the likelihood of dystonia. Due to the ambiguity surrounding the phenomenon, several researchers have adopted the phrase “dystonia-like [70]”. Other authors believe that due to the characteristic of occurring due to long-term therapy of antipsychotics, this phenomenon should be characterized as an extrapyramidal syndrome. On the other hand, some authors presuppose that this syndrome should be distinctly described as a separate movement disorder and should not be included in extrapyramidal symptoms or as a form of dystonia [27].

A relationship between the timeline of drug administration and the mechanism of action of the offending drugs associated with Pisa syndrome can be observed (Table 6). The direct effect of high-dose levodopa and monoamine oxidase inhibitors on dopamine receptors can rapidly cause an increase in the dopamine concentration in the synaptic membrane. In this context, Pisa syndrome can occur and subside faster when associated with these drugs. On the other hand, medications that indirectly affect the levels of levodopa, as in the case of istradefylline, could potentially lead to a relative delay in the onset of Pisa syndrome, which can also be observed in the recovery after the withdrawal of the offending agent.

## 8. Differential Diagnosis

While there are also secondary, symptomatic focal dystonias, idiopathic focal dystonias account for the majority of the differential diagnoses [165]. To accurately predict the disease course and prognosis, it is critical to differentiate between the idiopathic and symptomatic types. Unclear etiology, laboratory, and neuroimaging testing are generally not as helpful with adult-onset focal dystonia, which is thought to be idiopathic. Information on the clinical features, distribution, and onset is helpful. The most typical kind of symptomatic adult-onset focal dystonia is tardive dystonia, which can be elicited via the use of neuroleptic medications over a prolonged period of time, as well as by using antiemetic or vertigo medications [166]. Patients with tardive dystonia tend to respond to therapy less favorably than patients with idiopathic dystonia [167].

Patients with idiopathic cervical dystonia frequently describe a progressive development of symptoms [168]. In contrast to posttraumatic dystonia, which manifests in the exact body location as focal dystonia, cervical dystonia is usually not associated with recent trauma [169]. On the other hand, individuals who have post-traumatic dystonia have a substantial restriction in their range of motion, no antagonistic effect, and no recovery after sleeping [170].

The abnormal neurological finding in idiopathic dystonia is the presence of dystonic postures and movements. Degenerative parkinsonian disorders may be accompanied by abnormal spinal postures suggesting Pisa syndrome [171]. However, these patients will probably have additional neurological symptoms that lead to a primary diagnosis.

Any patient younger than 40 years old presenting with Pisa syndrome should be tested for Wilson’s disease. In this condition, different clinical forms of dystonia can be observed, and the treatment involves dietary changes and chelation therapy [172].

## 9. Management

### 9.1. Conservative Management

The condition generally disappears after the offending medications are discontinued. Although a pharmacological therapy for drug-induced Pisa syndrome has not been established, Suzuki et al. have reported that anticholinergic drugs are effective in about forty percent of patients who have episodes of Pisa syndrome, with the remaining patients responding to the withdrawal or reduction in daily doses of antipsychotic drugs [18]. Interestingly, the offending drug dose can be adjusted with improvement of the symptoms, but this should be assessed on a case-by-case basis. In some reports, a dose-dependent effect was observed [71].

Another important fact to mention is the independent relationship with drugs. There are case reports associating the switching of the offending drug with one inside the same class with improvements in Pisa syndrome. This should be mainly observed in cases where the patients have resistance to many antipsychotics and only respond to specific groups of medications.

Some authors reported a wash-out period between the discontinuation of the offending drug and the beginning of the following drug [74].

### 9.2. Specialized Peripheral Denervation

This surgical procedure retains the innervation of noncontributory muscles while denervating the muscles that cause aberrant movements [166]. Before considering surgery, most specialized centers recommend the stabilization of the clinical symptoms for at least one year. Additionally, individuals with minimal extension and pure rotatory torticollis have the best postoperative outcomes, while those with significant arthrosis or preexisting fibrosis are more likely to have less favorable outcomes [173].

### 9.3. Deep Brain Stimulation

Patients with Parkinson’s disease reported fewer instances of tremor after receiving targeted brain stimulation in their thalamus and basal ganglia [174,175]. As a result, stereotactic surgery was utilized to treat dystonia, but the mechanism for the improvement of motor symptoms with deep brain stimulation (DBS) is still unclear [176]. In individuals with intractable dystonias, DBS of the subthalamic nucleus (STN) or globus pallidus internus (GPi) has recently been performed [177].

Microelectrodes were inserted into the GPi during the procedure, usually bilaterally, with the GPi being identified and the microelectrode placement being guided by microstimulation [178]. Occasionally, electrode placement was guided by magnetic resonance imaging and microelectronic recording. After the procedure, several visits are needed to properly configure the stimulator’s settings [179]. The reversibility of the treatment, the flexibility in changing the stimulation parameters, and ongoing access to the therapeutic target are all benefits of DBS (Figure 4).

## 10. New Perspectives and Future Studies

Little is known regarding patients with Pisa syndrome, partially because there are no specific criteria for this disorder. A significant milestone for the continuous research of this disorder is the development of diagnostic criteria. In this manuscript, we proposed four criteria based on previous studies. However, there is a need for the development of a consensus among movement disorders specialists.

One interesting point of discussion concerns the prevalence of this disorder among different studies. There is a significant difference in the prevalence of drug-induced Pisa syndrome, from 0.037% to 9.3%. In this context, the studies did not clearly describe these movement disorders’ phenomenology. Moreover, experimental studies with animal models and perhaps neuropathological studies that explore this condition’s pathophysiology and molecular basis are also needed.

Pharmacoepidemiological studies should be conducted to assess this disorder’s prevalence among different medication classes. There are various opinions among specialists regarding the existence of Pisa syndrome secondary to drugs. Some movement disorder specialists do not agree with this association even though a significant number of reports are in the literature. The continuous publication of cases of Pisa syndrome associated with medications is recommended.

Some authors believe that electrodiagnostic studies do not provide criteria for supporting the diagnosis. They rely only on the clinical manifestation of this disorder. We believe that this approach is feasible in clinical practice, but severe cases or those with unreliable responses should be further investigated. There are a variety of causes for this disorder, and many other conditions can have similar presentations.

## 11. Limitations

Most reports are from single case studies, which can make controlling confounding variables difficult. Also, most studies were not reported by movement disorder specialists. These two critical drawbacks can lead to misinterpretations of the Pisa syndrome. Furthermore, the patients reported having variable comorbidities, and medications challenge the diagnosis for a single association. Therefore, the etiology of Pisa syndrome in these cases should be assumed as multifactorial, and specific algorithms, such as the Naranjo algorithm, should be used to determine the likelihood of whether an adverse drug reaction is actually due to the drug rather than the result of other factors.

## 12. Conclusions

In conclusion, the present study has examined the complex and varied phenomena of drug-induced Pisa syndrome, describing its clinical manifestation, underlying processes, and therapeutic approaches through a thorough review of the existing literature. Also, it is worth mentioning that special attention should be given to the administration of drugs that are known to exacerbate Pisa syndrome. Early detection and management are crucial to reduce the adverse effects on patients’ quality of life. We have also emphasized the importance of conducting more research to clarify the pathophysiological pathways behind drug-induced Pisa syndrome and to create more specialized and efficient therapeutic strategies.

## Figures and Tables

**Figure 1 geriatrics-09-00100-f001:**
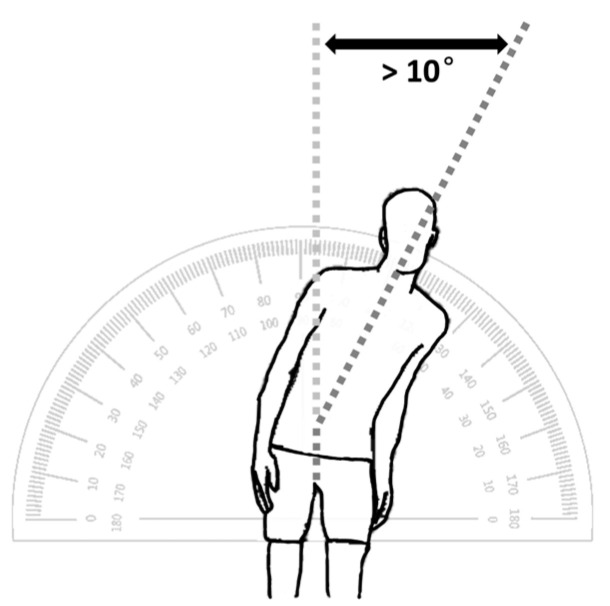
Pisa syndrome is characterized by more than ten degrees of constant lateral curvature of the spine when upright, without any evident rotation of the spinal bones, resembling the posture of the Leaning Tower of Pisa.

**Figure 2 geriatrics-09-00100-f002:**
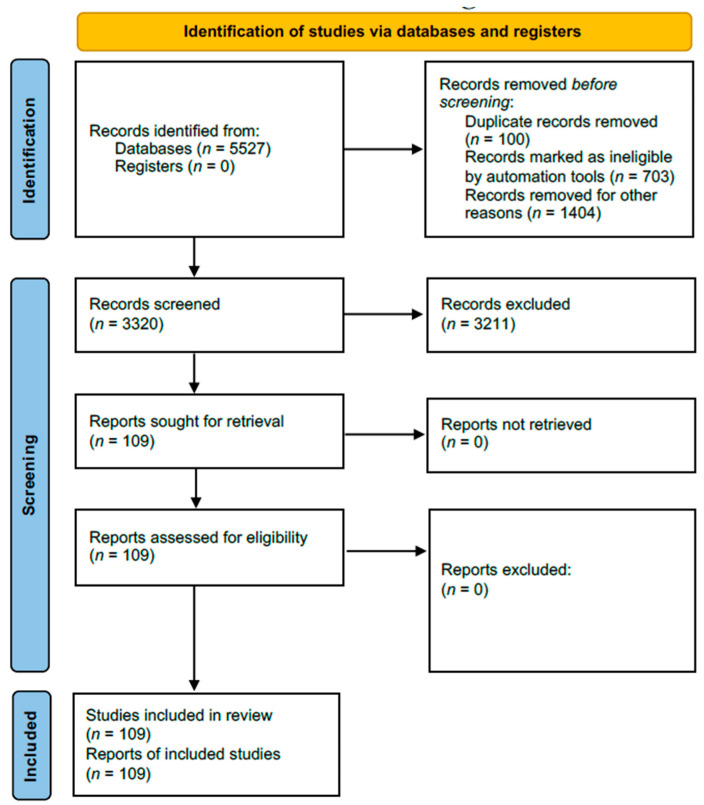
Flowchart of the screening process.

**Figure 4 geriatrics-09-00100-f004:**
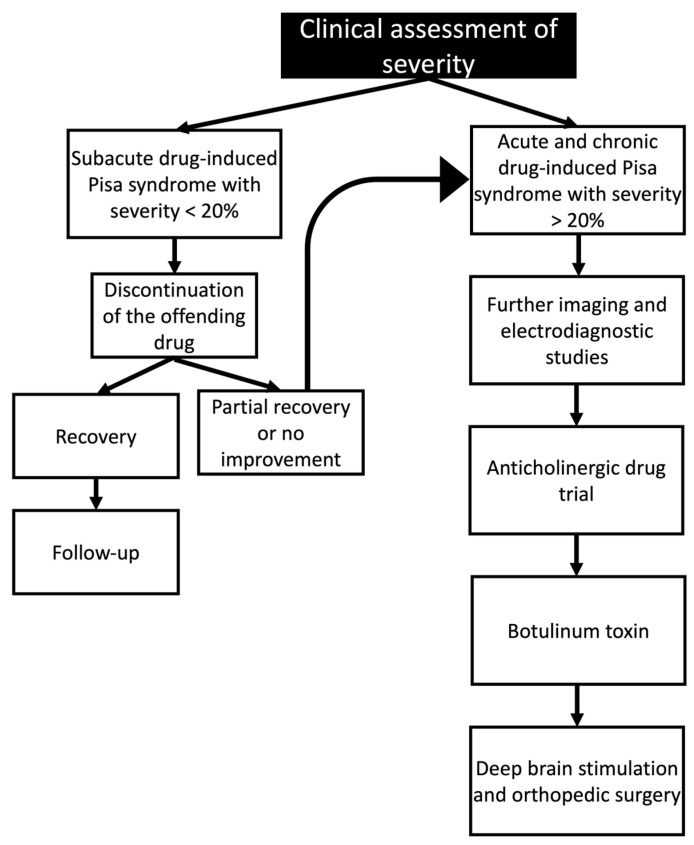
Clinical management of drug-induced Pisa syndrome based on the clinical assessment of the symptom’s severity.

**Table 1 geriatrics-09-00100-t001:** FreeText and MeSH search terms in the US National Library of Medicine.

Query	Search Terms	Results
Pisa syndrome	“Pisa” [All Fields] AND (“syndrom” [All Fields] OR “syndromal” [All Fields] OR “syndromally” [All Fields] OR “syndrome” [MeSH Terms] OR “syndrome” [All Fields] OR “syndromes” [All Fields] OR “syndromes” [All Fields] OR “syndromic” [All Fields] OR “syndroms” [All Fields])	3506
Pleurothotonus	“Pleurothotonus” [All Fields]	31

**Table 2 geriatrics-09-00100-t002:** Prevalence of drug-induced Pisa syndrome in different studies.

Reference	Prevalence	Note
Yassa et al. (1991) [12]	9.3% of females and 6.4% of males, and the mean was 8.3%	Antipsychotic treatment between 1984 and 1989 at a Canadian center
Stübner et al. (2000) [13]	0.037%	Large population with psychiatric disorders
Tinazzi et al. (2015) [14]	15%	Patients with PD who developed dopaminergic drug-induced Pisa syndrome
Lee et al. (2018) [15]	0.45% of males and 0.37% of females	Neuroleptic treatment. Also, there is a higher frequency of occurrence in males than in females unlike previous studies
Naguy et al. (2022) [16]	0.04%	Patients with Alzheimer’s disease taking acetylcholinesterase inhibitors
Wyant et al. (2023) [17]	0.09% to 1.7%	Antipsychotic treatment

**Table 3 geriatrics-09-00100-t003:** Literature review of the cases of Pisa syndrome secondary to drugs.

Reference	Age/Sex	Cause of Pisa Syndrome	Movement Disorder Onset	Management	Movement Disorder Recovery	Note
Ekbom et al. (1972) [3]	59/F	Methylperone	Days	Methylperone was discontinued.	2 months. Partial recovery	
	63/F	Methylperone	7 days	Methylperone was discontinued.	2 days. CR.	
	69/F	Methylperone	2 days	Methylperone was discontinued.	2 weeks	
Yassa et al. (1985) [35]	57/M	Chlorpromazine	8 years	Chlorpromazine was discontinued. Benztropine trial was unsuccessful.	16 h. CR.	
	32/M	Chlorpromazine	1 week	Benztropine trial was unsuccessful.	No recovery.	
Guy et al. (1986) [36]	44/F	Antipsychotic	NA	Antipsychotic was discontinued.	Undetermined time. Partial recovery.	
Saxena et al. (1986) [37]	37/M	Fluphenazine	3 years	Fluphenazine was discontinued.	6 months. CR.	
Amore et al. (1988) [38]	25/M	Haloperidol and chlorpromazine	NA	Haloperidol and chlorpromazine were discontinued.	Undetermined time. CR.	
Davidson et al. (1988) [39]	66/M	Haloperidol	1 week	Haloperidol dose was maintained. Amantadine was prescribed.	Undetermined time. Partial recovery.	
Remington et al. (1988) [40]	46/M	Amitriptyline and thioridazine.	4 days	Amitriptyline and thioridazine were discontinued.	2 days.	Benztropine was administered, but no effect was observed.
Chiu et al. (1989) [41]	38/F	Several antipsychotics	Variable times	Several antipsychotics.	Variable times. Partial recovery.	Patient had partial improvement even though structural changes were observed.
Turk et al. (1991) [42]	15/F	Trifluoperazine	4 days	Trifluoperazine was discontinued.	1 week. CR.	
Fichtner et al. (1992) [43]	15/F	Droperidol, prochlorperazine	2 days	Droperidol and prochlorperazine were discontinued.	1 day. CR.	Misdiagnosed as a functional disorder.
Suzuki et al. (1992) [44]	22/M	Haloperidol	NA	Haloperidol was switched to pimozide.	Undetermined time. CR.	
Suzuki et al. (1997) [33]	58/M	Nortriptyline	9 days	Nortriptyline dose was maintained. Biperiden was started.	4 days. CR.	
Bruneau et al. (1998) [45]	45/F	Clozapine	6 weeks	Clozapine dose was maintained. Procyclidine was started.	Undetermined time. CR.	
Padberg et al. (1998) [46]	85/F	Sertindole	10 weeks	Sertindole dose was reduced.	4 days. CR.	
Kwak et al. (2000) [47]	53/F	Donepezil	4 weeks	Donepezil was discontinued.	7 days	
	73/F	Rivastigmine	10 days	Rivastigmine was discontinued.	3 weeks	
Kropp et al. (2001) [48]	62/F	Metoclopramide	2 days	Metoclopramide was discontinued.	8 days. CR.	
Miyaoka et al. (2001) [49]	57/M	Donepezil	4 weeks	Donepezil was discontinued.	7 days	
Wölfl et al. (2001) [50]	33/M	Intoxication with tramadol, alprazolam, diazepam, bromazepam, alcohol and dothiepin	3 days	Intoxication episode.	3 days. CR.	
Jagadheesan et al. (2002) [51]	24/F	Risperidone	2 weeks	Risperidone was discontinued. Trihexyphenidyl was started.	Undetermined time. Partial recovery.	
Harada et al. (2002) [52]	25/M	Risperidone	15 months	Risperidone was discontinued. Biperiden dose was decreased.	3 weeks. Partial recovery.	
Azcano et al. (2003) [53]	19/M	Paliperidone	1 year	Paliperidone was discontinued.	Undetermined time. CR.	
Villarejo et al. (2003) [54]	77/F	Donepezil	4 weeks	Donepezil was discontinued. Biperiden was prescribed.	NA	
	72/F	Donepezil	2 months	Donepezil was discontinued.	10 days	
Ziegenbein et al. (2003) [55]	38/F	Ziprasidone	18 days	Ziprasidone was discontinued.	14 days. CR.	
Cossu et al. (2004) [34]	72/F	Galantamine	4 weeks	Galantamine was discontinued. Responsive to botulinum toxin.	Partial recovery.	
Duggal et al. (2004) [56]	82/F	Clozapine	11 days	Clozapine was discontinued.	2 days. CR.	
Cannas et al. (2005) [57]	56/M	Pergolide	4 months	Pergolide was discontinued.	3 months. CR.	
Vanacore et al. (2005) [58]	83/F	Donepezil	1 month	Donepezil was discontinued.	16 days	
	84/F	Donepezil	4 months	Donepezil was discontinued.	7 days	
	75/F	Donepezil	2.5 years	Donepezil was discontinued.	14 days	
Arora et al. (2006) [59]	22/M	Olanzapine	9 months	Olanzapine was switched to clozapine.	6 weeks. CR.	
González Pablos et al. (2006) [60]	60/F	Risperidone	1 year	Risperidone was discontinued.	NA.	
Yohanan et al. (2006) [61]	65/M	Valproate	Months	Valproate was discontinued.	Undetermined time. CR.	
Huvent-Grelle et al. (2007) [62]	78/F	Galantamine	11 months	Galantamine was discontinued.	15 days	
	75/F	Donepezil	4 years	Donepezil was discontinued.	15 days	
	72/F	Galantamine	6 months	Galantamine was discontinued.	NA	Switching to rivastigmine.
	76/F	Donepezil	12 months	Donepezil was discontinued.	Partial recovery.	
	79/M	Rivastigmine	24 months	Rivastigmine was discontinued.	NA	
Hung et al. (2007) [63]	39/F	Clozapine	5 months	Clozapine dose was reduced.	4 weeks. CR.	
Nishimura et al. (2007) [64]	29/M	Risperidone	4 months	Risperidone was discontinued. Trihexyphenidyl was started.	2 months. Partial recovery.	
Rota et al. (2007) [65]	77/F	Aripiprazole	6 days	Aripiprazole was discontinued.	3 days	
Strauss et al. (2007) [66]	50/F	Risperidone	NA	Risperidone was discontinued.	Undetermined time. CR.	
Chen et al. (2008) [67]	65/F	Galantamine	6 months	Galantamine was discontinued.	Partial recovery.	
Cordeiro et al. (2008) [68]	18/M	Risperidone	8 weeks	Risperidone was discontinued. Biperiden was started.	3 days. CR.	
Ogihara et al. (2008) [69]	70/F	Donepezil	NA	Donepezil was discontinued.	4 weeks.	Pisa syndrome only appeared after the introduction in a long-term donepezil therapy.
	67/F	Donepezil	16 months	Donepezil was discontinued.	3 weeks	Previous episodes of Pisa syndrome with quetiapine and trazodone.
Solla et al. (2008) [70]	62/M	Levodopa/carbidopa/entacapone	2 weeks	Levodopa/carbidopa/entacapone was discontinued.	Days. CR.	
Uemura et al. (2008) [71]	56/F	Clomipramine	2 months	Clomipramine dose was reduced.	1 month. CR.	
Wang et al. (2008) [72]	77/M	Rivastigmine	2 weeks	Rivastigmine dose was reduced.	3 days. CR.	Rivastigmine was correlated with a dose-dependent effect.
Huvent-Grelle et al. (2009) [73]	98/M	Donepezil	3 months	Donepezil was discontinued.	7 days. CR.	
	80/M	Donepezil	18 months	Donepezil was discontinued.	Partial recovery	
Huang et al. (2009) [74]	46/F	Aripiprazole	3 days	Aripiprazole was discontinued.	3 days	
Kuo et al. (2009) [75]	57/M	Amilsupride	10 days	Amilsupride was discontinued.	2 days	
Silić et al. (2009) [76]	NA/M	Ziprasidone	6 days	Ziprasidone was discontinued. Biperiden was started.	1 day	
Yeh et al. (2009) [77]	38/M	Ziprasidone	1 month	Ziprasidone was discontinued. Amilsupride was started without motor symptoms.	1 week	
Walder et al. (2009) [27]	69/F	Quetiapine	5 days	Quetiapine was discontinued.	3 days	
Albuquerque et al. (2010) [78]	33/F	Haloperidol	13 year	Haloperidol was discontinued. Trihexyphenidyl was started.	3 months. Partial recovery.	New worsening with risperidone.
Chen et al. (2010) [79]	44/F	Aripiprazole	2 months	Aripiprazole was discontinued.	4 days	
Li et al. (2010) [80]	26/M	Amilsupride	2 weeks	Amilsupride was discontinued.	12 days. Partial recovery.	
Ulhaq et al. (2010) [81]	31/M	Risperidone	NA	Risperidone was discontinued.	Undetermined time. CR.	Reports of patients with intellectual disability.
	46/F	Risperidone	NA	Risperidone was discontinued.	Undetermined time. CR.	
Wu et al. (2010) [82]	29/M	Amisulpride	3 days	Amilsupride was discontinued. Trihexyphenidyl was started.	1 day. CR.	
Fasano et al. (2011) [5]	64/M	Rasagiline	3 weeks	Rasagiline was discontinued.	4 weeks	
	73/M	Rasagiline	4 weeks	Rasagiline was discontinued.	4 weeks	
	72/M	Rasagiline	3 weeks	Rasagiline was discontinued.	2 weeks	
	67/F	Rasagiline	4 weeks	Rasagiline was discontinued.	3 weeks	
Miodownik et al. (2011)—Case 1 and Case 2 [83]	58/M	Sertindole	6 weeks	Sertindole was discontinued.	2 weeks. Partial recovery.	
	38/M	Ziprasidone	3 months	Ziprasidone was discontinued.	2 weeks. Partial recovery.	
Shinfuku et al. (2011) [84]	71/F	Donepezil	12 months	Donepezil was discontinued.	Partial recovery.	
Ducasse et al. (2012) [85]	72/F	Risperidone	1 weeks	Risperidone was discontinued.	Undetermined time. Partial recovery.	
Ioannidis et al. (2012) [86]	74/F	Donepezil	1 day	Donepezil was discontinued.	8 days	
Kaufmann et al. (2012) [87]	23/F	Risperidone	1 year	Risperidone was discontinued. Biperiden was discontinued. Baclofen was started.	1 week. Partial recovery.	
Leelavathi et al. (2012) [88]	80/M	Rivastigmine	18 months	Rivastigmine was discontinued.	Partial recovery.	Switching to donepezil.
Liuu et al. (2012) [89]	73/M	Valproate	NA	NA	NA	
Perrone et al. (2012) [90]	47/F	Sertraline	1 month	Sertraline was discontinued.	3 weeks. Partial recovery.	
Iuppa et al. (2013) [91]	31/M	Risperidone	11 years	Risperidone was discontinued.	1 week	
Méndez Guerrero et al. (2013) [92]	61/F	Mirtazapine	Single dose	Mirtazapine was discontinued.	Three days. CR.	
Teng et al. (2013) [93]	42/M	Paliperidone	2 months	Paliperidone was discontinued.	2 weeks. CR.	
Galati et al. (2014) [94]	68/F	Ropinirole	NA	Ropinirole was discontinued.	Undetermined time. CR.	
Olsson et al. (2014) Case 2 [95]	38/M	Olanzapine	days	Olanzapine dose was reduced.	Undetermined time. CR.	
Pan et al. (2014) [96]	18/M	Paliperidone	12 months	Paliperidone was discontinued.	1 week. CR.	
Wang et al. (2014) [97]	60/F	Clotiapine withdrawal	3 days	Baclofen and biperiden were started.	Partial recovery.	
Zannas et al. (2014) [98]	74.7 (mean)/17F+ 4M	Donepezil	NA	NA	NA	Food and Drug Administration Adverse Event Reporting System database
	73.9 (mean)/13F + 4M	Galantamine	NA	NA	NA	
	75.1 (mean)/5F + 9M	Rivastigmine	NA	NA	NA	
Arjunan et al. (2015) [99]	83/M	Risperidone	2 days	Risperidone was discontinued.	Undetermined time. CR.	
Faridhosseini et al. (2015) [2]	33/F	Clozapine	2 years	Clozapine dose was reduced. Biperiden was started.	1 month. CR.	
Miletić et al. (2015) [100]	67/F	Risperidone	3 days	Risperidone was discontinued.	2 weeks. Partial recovery.	
Pellene et al. (2015) [20]	71/M	Pramipexole	4 years	Pramipexole was discontinued.	20 days. CR.	
Botturi et al. (2016) [101]	59/M	Valproate	3 months	Valproate was discontinued.	Weeks. Partial recovery.	
De Risio et al. (2016) [102]	25/F	Aripiprazole	28 months	Aripiprazole was discontinued.	4 weeks	
Güneş et al. (2016) [28]	15/M	Risperidone	4 years	Risperidone dose was reduced.	2 weeks	
Tsou et al. (2016) [103]	37/M	Paliperidone	3 months	Paliperidone was discontinued. Anticholinergic drugs were prescribed, but no improvement was observed.	1 month	
Hsu et al. (2017) [104]	57/F	Rivastigmine	19 months	Rivastigmine was discontinued.	1 month	
Kumar et al. (2017) [105]	49/M	Lithium	2 years	Lithium was discontinued.	3 months	
Mahmoud et al. (2017) [106]	14/NA	Valproate	5 years	Valproate was discontinued.	NA	
Pollock et al. (2017) [107]	87/M	Donepezil	2 years	Donepezil was discontinued.	3 months	
Suresh Kumar et al. (2017) [108]	52/F	Clozapine	2 years	NA	NA	
Sutter et al. (2017) [109]	56/F	Risperidone	4 years	Risperidone was discontinued.	6 months	
Chao et al. (2018) [110]	67/F	Rivastigmine	5 days	Rivastigmine was discontinued.	7 days	
Filipe et al. (2018) [111]	56/F	Clozapine and mirabegron	5 days	Clozapine and mirabegron	30 days. CR.	
Huang et al. (2018) [112]	31/F	Clotiapine	2 weeks	Clotiapine was discontinued.	10 days	
Lee et al. (2018) [15]	47 (mean)/9M, 4F	Antipsychotics	NA	NA	NA	13 individuals
López-Blanco et al. (2018) [113]	76/F	Betahistine	Single dose	Betahistine was discontinued.	1 day	
Mukku et al. (2018) [114]	60/M	Donepezil	3 months	Donepezil was discontinued. Promethazine was prescribed with partial response. Clonazepam was infective.	3 days	
Sosa et al. (2018) [115]	77/F	Codeine	1 day	Codeine was discontinued.	3 days	
Yamada et al. (2018) [116]	79/F	Mirtazapine	12 days	Mirtazapine was discontinued.	3 days	
Yasuda et al. (2018) [117]	68/M	Istradefylline	4 months	Istradefylline was discontinued.	4 months	Patient already had a previous trunk deviation due to lumbar spondylosis. The istradefylline increased the angle.
Ciner et al. (2019) [118]	45/M	Paliperidone	2 years	Paliperidone was discontinued.	NA	
Guler et al. (2020) [119]	12/F	Olanzapine	3 months	Olanzapine was discontinued. Biperiden was prescribed.	15 days	
Hsieh et al. (2020) [120]	62/M	Lithium	1 week	Lithium was discontinued.	1 day	There was worsening of the angle with lithium therapy. Dose-dependent effect.
Mimura et al. (2020) [1]	60/F	Galantamine	2 weeks	Galantamine was discontinued.	2 weeks	
Atmaca et al. (2021) [121]	81/F	Piribedil	1 month	Piribedil was discontinued.	1 month	
Bicho et al. (2021) [122]	48/F	Antipsychotic	NA	Antipsychotic discontinuation. Anticholinergic prescribed.	NA	
Bruggeman et al. (2021) [123]	68/F	Donepezil	5 years	Donepezil discontinuation.	6 months. CR.	Video.
Erdem et al. (2021) [124]	47/M	Amilsupride	10 years	Amilsupride discontinued. Biperiden and baclofen were started.	Partial recovery	Attempt of botulinum toxin injection without improvement.
Mascia et al. (2021) [31]	46/M	Cocaine, sniffed	1 h	Biperiden prescription.	2 h	Cocaine rechallenged revealed reappearance of the symptoms.
Santos et al. (2021) [125]	73/F	Clozapine	NA	NA	NA	
Simões et al. (2021) [126]	72/F	Donepezil	5 months	Donepezil discontinuation.	7 days. CR.	Rivastigmine was attempted, but she had similar symptoms of donepezil use.
Nagai et al. (2022) [127]	71/F	Brexpiprazole	2 months	Brexpiprazole was switched to quetiapine.	NA. Partial recovery.	
Naguy et al. (2022) [16]	13/M	Lithium	4 weeks	Lithium discontinuation.	2 weeks. CR.	
Shen et al. (2022) [128]	62/M	Clozapine	3 years	Clozapine dose was reduced.	3 days. Partial recovery.	Pisa syndrome with oral-buccal-lingual dyskinesia.
Weise et al. (2022) [129]	41/F	Clozapine and cariprazine	4 weeks	Discontinuation of cariprazine. Lorazepam and biperiden were prescribed.	3 days. CR.	Add-on cariprazine in resistant schizophrenia.
	63/M	Clozapine and cariprazine	2 weeks	Discontinuation of cariprazine. Prescription of biperiden.	3 weeks. CR.	Add-on cariprazine in resistant schizophrenia.
Muller et al. (2023) [130]	NA	Clozapine	NA	NA	NA	
Waykar et al. (2023) [131]	75/F	Donepezil	6 months	Discontinuation of donepezil. Mematine prescription.	1 month. CR.	

Abbreviations: CR, complete recovery; F, female; M, male; NA, not available/not applicable.

**Table 4 geriatrics-09-00100-t004:** Drug-induced Pisa syndrome, according to the drug classes.

Drug Class	Number of Reports	Medications
Acetylcholinesterase inhibitors	87	Donepezil [126], galantamine [1], rasagiline [4], rivastigmine [47].
Atypical antipsychotics	50	Amilsupride [124], aripiprazole [102], brexpiprazole [127], clotiapine [112], clozapine [129], olanzapine [119], paliperidone [118], risperidone [109], ziprasidone [83].
Typical antipsychotics	30	Chlorpromazine [35], droperidol [43], fluphenazine [37], haloperidol [78], methylperone [3], sertindole [46], trifluoperazine [42].
Antiparkisonian medications	6	Istradefylline [117], levodopa/carbidopa/entacapone [70], pergolide [57], piribedil [121], pramipexole [20], ropinirole [94].
Anti-seizure medications	4	Valproate [106].
Antidepressants	3	Mirtazapine [116], sertraline [90].
Mood stabilizers	3	Lithium [16].
Tricyclic antidepressants	3	Amitriptyline [40], clomipramine [71], nortriptyline [33].
Opioid	1	Codeine [115].
Antiemetics	1	Metoclopramide [48].
Addictive stimulant drug	1	Cocaine [31].
Anti-vertigo medication	1	Betahistine [113].
Intoxication	1	Intoxication with a mixture of tramadol, alprazolam, diazepam, bromazepam, alcohol, and dothiepin [50].

**Table 5 geriatrics-09-00100-t005:** Diagnostic criteria for Pisa syndrome by Rissardo et al.

Criteria	Reference
(A) More than 10 degrees of lateral trunk flexion. Bonanni et al. recommended more than 15 degrees. Tinazzi et al. classified mild (<20°) or severe (>20°) according to the angle.	Doherty et al. (2011) [136] Bonanni et al. (2007) [135] Tinazzi et al. (2015) [14]
(B) Lateral trunk flexion is relieved by passive mobilization or supine posture.	Doherty et al. (2011) [136]
(C) Lateral trunk flexion worsens during walking and in the absence of any mechanical restriction of the trunk.	Bonanni et al. (2007) [131] Ekbom et al. (1972) [3]
(D) Electromyography pattern of activation of paraspinal muscles and non-paraspinal muscles.	Tassorelli et al. (2012) [133]
(E) Supplementary imaging of neurospine for evaluation of degenerative process and musculoskeletal abnormalities.	

**Table 6 geriatrics-09-00100-t006:** The time between movement disorder onset and recovery of different classes of drugs associated with Pisa syndrome.

Medication	Movement Disorder Onset	Movement Disorder Recovery	Reference
Levodopa/Benserazide	15 days	20 days	Cannas et al. (2009) [165]
Carbidopa/Levodopa/Entacapone	2 weeks to 1 month	Few days to 10 days	Solla et al. (2008) [70] Cannas et al. (2009) [165]
Rasagiline	3–4 weeks	2–4 weeks	Fasano et al. (2011) [4]
Pramipexole	2 months	40 days	Cannas et al. (2009) [165]
Levodopa/Carbidopa	2 months	2 months	Cannas et al. (2009) [165]
Pergolide	2–3 months	3 months	Cannas et al. (2005) [57] Cannas et al. (2009) [165]
Istradefylline	4 months	4 months	Yasuda et al. (2018) [117]
Ropinirole	1 year	3 months	Galati et al. (2014) [94]

## Data Availability

No new data was created.

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
