# Peer review of "Pisa Syndrome Secondary to Drugs: A Scope Review"

_geriatrics, 2024, doi:10.3390/geriatrics9040100_

Round 1

Reviewer 1 Report

Comments and Suggestions for Authors

better design is systemiatic research following PRISMA checklist, please follow it in your research

Author Response

Reviewer: better design is systemiatic research following PRISMA checklist, please follow it in your research

Authors: Dear Reviewer, We agree with your comment and wrote a specific chapter about the study's methodology. We followed the PRISMA checklist and guidelines. We believe that your comment significantly improved the quality of the manuscript.

Reviewer 2 Report

Comments and Suggestions for Authors

The authors cover the body of literature on a rare drug induced postural deformity commonly known as Pisa Syndrome. The article as it stands is not suitable for publication. 

I recommend elucidating better the pathological mechanisms underlying this condition

I recommend the authors create a materials and methods section explaining how they reviewed the literature. Saying that you have comprehensively reviewed the literature means nothing: have you performed a systematic, scoping or narrative review? Was this review registered in Prospero or other repositories? Have you adhered to the appropriate checklist for reporting of the results of a narrative or scoping or systematic review?

- I recommend downsizing the reference list to a maximum of 100 references. Many articles included in the current reference list are redundant, many of them are from the same authors or their group. At present this is not acceptable. 

The authors need to conduct a major restructuring of their article if they wish to resubmit and see it published. 

Author Response

Reviewer: The authors cover the body of literature on a rare drug induced postural deformity commonly known as Pisa Syndrome. The article as it stands is not suitable for publication. I recommend elucidating better the pathological mechanisms underlying this condition.

Authors: We appreciate the reviewer's comment. Pisa syndrome is uncommonly seen in clinical practice, and in this review, we just evaluated Pisa syndrome as likely secondary to medications. We provided the most detailed explanation found in the literature regarding the pathophysiology of Pisa syndrome (Figure 2). For a comparison with our manuscript, we would like to ask the reviewer to read the article published in Lancet Neurology in 2016, in which the authors vaguely described pathophysiology. Noteworthy, little development was done in idiopathic Pisa syndrome in the last decade.

Barone P, Santangelo G, Amboni M, Pellecchia MT, Vitale C. Pisa syndrome in Parkinson's disease and parkinsonism: clinical features, pathophysiology, and treatment. Lancet Neurol. 2016 Sep;15(10):1063-74. doi: 10.1016/S1474-4422(16)30173-9. Epub 2016 Aug 8. PMID: 27571158.

Reviewer: I recommend the authors create a materials and methods section explaining how they reviewed the literature. Saying that you have comprehensively reviewed the literature means nothing: have you performed a systematic, scoping or narrative review? Was this review registered in Prospero or other repositories? Have you adhered to the appropriate checklist for reporting of the results of a narrative or scoping or systematic review?

Authors: We initially believed that the methodology did not need to be described because we included all the cases from the literature from six databases. But, after the reviewer's comment, we believed that it was unclear, and we provided a detailed description of the etiology, including search strategy, inclusion and exclusion criteria, data extraction, statistical analysis, and definitions. We would like to thank you, the reviewer because this comment significantly improved the quality of the manuscript.

Reviewer: I recommend downsizing the reference list to a maximum of 100 references. Many articles included in the current reference list are redundant, many of them are from the same authors or their group. At present this is not acceptable. The authors need to conduct a major restructuring of their article if they wish to resubmit and see it published. 

Authors: We understand the Reviewer's comment about the number of references, but we would like to break down the number of references for better understanding. There are 109 references from which the initial statistical study results were obtained. Five references for the introduction. Eight references were for epidemiology. Ten references to diagnostic criteria. There were 25 references for pathophysiology, in which basic studies were included to explain the specific pathophysiology of drug-induced Pisa. Seven were used for the clinical manifestations. 6 references were used for the differential diagnosis. Nine references for the management.

We believe that the current references are needed to explain the main idea of the manuscript. It is worth mentioning that this would be the most extensive review in the literature regarding this specific subject; there is no other review with a similar idea.

Reviewer 3 Report

Comments and Suggestions for Authors

This scoping review aims to examine drug-induced Pisa syndrome, including its clinical presentation, duration, diagnostic assessments, and treatments. This review provides a comprehensive and thorough examination of Pisa syndrome, covering its prevalence, associated risk factors, drugs linked to its development, physical examination and diagnostic criteria, pathophysiology, clinical manifestations, differential diagnosis, management strategies, new perspectives, and directions for future studies. The manuscript is well written.

At line 377, the authors state that “Some authors consider Pisa syndrome as a subtype of dystonia.” Please add references to support this statement.

Author Response

Reviewer: This scoping review aims to examine drug-induced Pisa syndrome, including its clinical presentation, duration, diagnostic assessments, and treatments. This review provides a comprehensive and thorough examination of Pisa syndrome, covering its prevalence, associated risk factors, drugs linked to its development, physical examination and diagnostic criteria, pathophysiology, clinical manifestations, differential diagnosis, management strategies, new perspectives, and directions for future studies. The manuscript is well written.

In line 377, the authors state that “Some authors consider Pisa syndrome as a subtype of dystonia.” Please add references to support this statement.

Authors:

Dear reviewer, we appreciate your comment, and a specific reference was described for the statement.

"Some authors consider Pisa syndrome as a subtype of dystonia [25]."

Please also note that the methodology of the manuscript was improved, and PRISMA guidelines were followed.

Round 2

Reviewer 1 Report

Comments and Suggestions for Authors

Now manuscript iis better

in methodology starta explaining you followed PRISMA 2020 checklist and refer it: https://www.bmj.com/content/372/bmj.n160

use flow chart prisma model: https://www.prisma-statement.org/prisma-2020-flow-diagram

add quality evaluation of included studies

what statistic analysis did you performed? I dont see anything

you have add a discusison section, perhaps text from each section is better there and add results section with tables and brief comments

Author Response

Now manuscript iis better

in methodology starta explaining you followed PRISMA 2020 checklist and refer it: https://www.bmj.com/content/372/bmj.n160

Authors: We agree with the reviewer, and we included the reference PRISMA 2020, and that we followed the guidelines.

use flow chart prisma model: https://www.prisma-statement.org/prisma-2020-flow-diagram

Authors: We agree with the reviewer and we included the PRISMA flowchart.

add quality evaluation of included studies

Authors: We kindly disagree with the quality evaluation for every study. Most are case reports and case series, and we described this significant drawback of the study in the limitations.

what statistic analysis did you performed? I dont see anything

Authors: We calculated the mean, standard deviation, and range using Excel. We described this in the statistical section of the methodology. 

you have add a discusison section, perhaps text from each section is better there and add results section with tables and brief comments

Authors: We appreciate this comment but have already reconfigured the manuscript because of another reviewer. Also, there's not too much specific data to do a specific result section. Therefore, the results were distributed throughout the manuscript.

--------------------------

Dear Reviewer, we would like to make a further comment. The present manuscript is a review based on previous case reports and case series due to the rarity of the disease. We included the PRISMA guideline as you requested, but if we carefully look at the PRISMA guideline, this review is not considered there. Otherwise, all the data is undefined or unclear. Currently, there are no specific systematic guidelines for manuscripts like the present. Even though there is a significant urge for a manuscript like this.

Previous reviews from the literature on Pisa syndrome did not follow any systematic approach and included idiopathic and drug-induced cases of Pisa syndrome as a unique pathology. Or pharmacovigilance studies that likely included other spinal pathologies with Pisa syndrome because they do not have clear data. Even for this review, there were cases in the literature describing camptocormia as a Pisa syndrome, through more than 10 cases in PubMed, which we excluded after reading the case and seeing the images or videos.

Here, we provide some basic statistics of the findings from these rare studies. And we believe that they are important to the literature since little is known about drug-induced Pisa syndrome. To be more specific, it is possible that all cases of Pisa syndrome that we see today, except for some extremely rare cases of Parkinson's plus forms, are related to medication.

We would like to ask the reviewer to read the closest comparison to our manuscript is a 2002 review about the same subject Published on CNS Drugs.

Suzuki T, Matsuzaka H. Drug-induced Pisa syndrome (pleurothotonus): epidemiology and management. CNS Drugs. 2002;16(3):165-74. doi: 10.2165/00023210-200216030-00003. PMID: 11888337.

--------------------------

We appreciate the Reviewer's taking the time to read and improve the quality of the manuscript.

Thank you very much.

Reviewer 2 Report

Comments and Suggestions for Authors

authors revised

Author Response

Dear Reviewer,
We appreciate all your effort in reviewing the manuscript and improving its quality.
Thank you very much!

Round 3

Reviewer 1 Report

Comments and Suggestions for Authors

paper is now ok for pubblication